# ML4C: Seeing Causality Through Latent Vicinity

## Abstract

Supervised Causal Learning (SCL) aims to learn causal relations from observational data by accessing previously seen datasets associated with ground truth causal relations. This paper presents a first attempt at addressing a fundamental question: *What are the benefits from supervision and how does it benefit?* Starting from seeing that SCL is not better than random guessing if the learning target is non-identifiable a priori, we propose a two-phase paradigm for SCL by explicitly considering structure identifiability. Following this paradigm, we tackle the problem of SCL on discrete data and propose ML4C. The core of ML4C is a binary classifier with a novel learning target: it classifies whether an Unshielded Triple (UT) is a v-structure or not. Starting from an input dataset with the corresponding skeleton provided, ML4C orients each UT once it is classified as a v-structure. These v-structures are together used to construct the final output. To address the fundamental question of SCL, we propose a principled method for ML4C featurization: we exploit the vicinity of a given UT (i.e., the neighbors of UT in skeleton), and derive features by considering the conditional dependencies and structural entanglement within the vicinity. We further prove that ML4C is asymptotically perfect. Last but foremost, thorough experiments conducted on benchmark datasets demonstrate that ML4C remarkably outperforms other state-of-the-art algorithms in terms of accuracy, robustness and transferability. In summary, ML4C shows promising results on validating the effectiveness of supervision for causal learning.

## 1  Introduction

The problem of causal learning is to learn causal relations from observational data [13]. The learned causal relations are typically represented in the form of a Directed Acyclic Graph (DAG), where each edge in the DAG indicates direct cause-effect relation between the parent node and child node.

The methods of causal learning mostly fall into four categories: constraint-based, score-based, continuous optimization method and functional causal models. Each of these methods takes a given dataset as input and outputs a DAG but with different criteria. For instance, the DAG should be consistent with conditional independencies in the data (constraint-based); or it is optimal w.r.t. a pre-defined score function under either combinatorial constraint (score-based) or continuous equality constraint (continuous optimization). In a nutshell, these methods can be viewed as *unsupervised* since they do not access additional datasets associated with ground truth causal relations.

A new line of research called *Supervised* Causal Learning (SCL), on the other hand, aims to learn causal relations in the supervised fashion: the algorithm has access to datasets associated with ground truth causal relations, in the hope that such supervision is beneficial to learning causal relations on newly unseen datasets. Despite several existing works on this direction (see Related Work), a fundamental question remains unanswered: *How is supervised causal learning possible*?

Submitted to 35th Conference on Neural Information Processing Systems (NeurIPS 2021). Do not distribute.

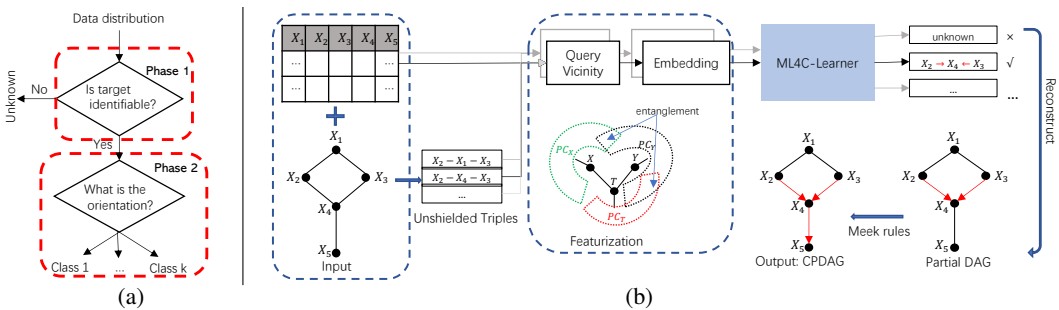

Figure 1: (a) Two-phase paradigm for supervised causal learning. (b) ML4C's workflow.

Specifically, compared with unsupervised causal learning methods, can we gain additional benefits from supervision? If the answer is positive, then what are the benefits and how does it benefit?

We tackle the problem by first seeing crucial connection between SCL and causal structure identifiability. Considering the problem of causal learning on discrete data, theorem in [24] states that, under standard assumptions (i.e., Markov assumption, faithfulness and causal sufficiency), we can only identify a graph up to its Markov equivalence class. Markov equivalence class is the set of DAGs having same skeleton and same v-structures, which can be represented by CPDAG (Completed Partially Directed Acyclic Graph). Thus, the (un)directed edges in the CPDAG indicate (non-)identifiable causal relations. Each non-identifiable edge in CPDAG can be oriented by either direction to equivalently fit the observational data. Given an SCL algorithm with learning target as the orientation of an edge, we see that it is not better than random guessing (or could be worse due to sample bias in training data) to predict any non-identifiable edge since we can assign either $X \rightarrow Y$ or $X \leftarrow Y$ with same input dataset. This statement is applicable to general learning target since an SCL algorithm can take different target such as orientation of an edge, the whole DAG, or others.

**Proposition 1.** *If the learning target is non-identifiable (i.e., every edge in the target is non-identifiable) a priori, then SCL is not better than random guessing.*

Consequently, we propose and advocate a two-phase paradigm for SCL, as depicted in Figure 1(a): phase one corresponds to a binary classification task, where an SCL algorithm needs to classify whether a specific learning target is identifiable or not; only if it is classified as identifiable, then we go to phase two to classify the specific orientation of the learning target. Following this paradigm, we tackle the problem of SCL on discrete data and propose an algorithm ML4C. The core of ML4C is a binary classifier with a novel learning target: it classifies whether an Unshielded Triple (UT: a triple of variables $\langle X, T, Y \rangle$ where $X$ and $Y$ are adjacent to $T$ but are not adjacent to each other) is a v-structure or not. Starting from an input dataset with the corresponding skeleton provided, ML4C orients each UT once it is classified as a v-structure. These v-structures are further used to construct a CPDAG as output. Such a single classifier facilitates both learning tasks in the two phases, since an identifiable UT implies that it is a v-structure [32] (i.e., up to the partial DAG before applying Meek rules [23] which is a standard post processing).

To address the fundamental question of SCL, we propose a principled method for ML4C featurization. Specifically, we exploit the *vicinity* of a given UT (i.e., the neighbors of UT in skeleton), and derive features by considering the conditional *dependencies* and structural *entanglement* within the vicinity. We further define discriminative predicate (i.e., a binary predicate function with domain as ML4C's feature set) and prove that there exist weak discriminative predicates and strong discriminative predicates (i.e., values of the predicates are one-to-one correspondence with ground truth labels). We further prove that ML4C is asymptotically perfect. Last but foremost, thorough experiments on benchmark datasets demonstrate that ML4C remarkably outperforms other state-of-the-art algorithms w.r.t. accuracy, robustness and transferability. Our main contributions are summarized as follows:

1. We advocate the two-phase paradigm for SCL with consideration of causal structure identifiability.

2. We propose an SCL algorithm ML4C, with the following novelties: i) **Learning Target**: The core of ML4C is a binary classifier with the orientation of a UT as its learning target to address the two-phase tasks simultaneously. ii) **Featurization**: A principled method to exploit vicinity information in terms of dependencies and entanglement of a given UT. iii) **Learnability**: We prove that ML4C

79   is asymptotically perfect. iv) **Empirical Performance**: Experiments conducted on benchmark
80   datasets demonstrate that ML4C remarkably outperforms other state-of-the-art algorithms.

## 2   Related Work

82   We divide literature on causal learning into supervised and unsupervised approaches, depending on
83   whether additional datasets (associated with ground truth causal relations) are accessed (supervised)
84   or not (unsupervised). In the literature of unsupervised causal learning, constraint-based methods aim
85   to identify a DAG which is consistent with conditional independencies. The learning procedure of
86   constraint-based methods first identifies the corresponding skeleton and then conducts orientation
87   based on v-structure identification [34]. The typical algorithm is PC [31], and there are also PC-
88   derived algorithms such as Conservative-PC [26], PC-stable [7] and Consistent-PC [19] which
89   improve the robustness on v-structure identification. Score-based methods aim to find the DAG which
90   is optimal w.r.t. a pre-defined score function under combinatorial constraint by a specific search
91   procedure, such as forward-backward search GES [6], hill-climbing [16], integer programming [8], or
92   by approximate algorithms based on order search [27]. Continuous optimization methods transform
93   the discrete search procedure into continuous equality constraint: NOTEARS [36] formulates the
94   acyclic constraint as a continuous equality constraint, it is further extended by DAG-GNN [35] to
95   support learning non-linear causal relations.

96   SCL emerges from the task of orienting edge in the continuous, non-linear bivariate case under
97   Functional Causal Model (FCM) formalism. Given a collection of cause-effect samples (dataset
98   $\sim$ binary label indicating whether $X \rightarrow Y$ or $X \leftarrow Y$), supervised approaches such as RCC [20],
99   NCC [21], D2C [4] and Jarfo [12] achieve better performance on predicting pairwise relations (i.e.,
100  orientation of an edge) than unsupervised approaches such as ANM [14] or IGCI [15]. Differently,
101  [18] sets the learning target as the whole DAG structure instead of pairwise relation and it is applied on
102  data which is generated by linear Structural Equation Model (SEM). We summarize the differences in
103  problem space between ML4C and the other SCL approaches as follows: i) We advocate a two-phase
104  learning paradigm and emphasize the relationship between identifiability and learnability. Specifically,
105  presuming additive noise model [14] or linear SEM with non-Gaussian noise [29] provides license
106  to identifiability thus the aforementioned approaches can be viewed as specific tasks in phase two.
107  ii) We tackle SCL's learnability not only via empirical evaluation but also by theoretical analysis to
108  shed light on the fundamental question of learnability. iii) ML4C deals with discrete data while other
109  approaches mainly focus on continuous data.

## 3   Background

### 3.1   Basic Notations

112  A discrete dataset $D_i$ consists of $n_i$ records and $d_i$ categorical columns, which represents $n_i$ instances
113  drawn i.i.d. from $d_i$ discrete variables $X_1, X_2, \cdots, X_{d_i}$ by a joint probability distribution $P_i$, which
114  is entailed by an underlying data generating process, denoted as DAG $G_i$.

115  **Markov factorization property**: Given a joint probability distribution $P$ and a DAG $G$, $P$ is said to
116  satisfy Markov factorization property w.r.t. $G$ if $P := P(X_1, X_2, \cdots, X_d) = \prod_{i=1}^{d} P(X_i|\mathrm{pa}_i^G)$,
117  where $\mathrm{pa}_i^G$ is the parent set of $X_i$ in $G$.

118  **Markov assumption**: $P$ is said to satisfy Markov assumption (or Markovian) w.r.t. a DAG $G$ if
119  $X \perp_G Y | Z \Rightarrow X \perp Y | Z$. Here $\perp_G$ denotes d-separation, and $\perp$ denotes statistical independence.
120  Markov assumption indicates that any d-separation in graph $G$ implies conditional independence in
121  distribution $P$. Markov assumption is equivalent to Markov factorization property [17].

122  **Faithfulness**: Distribution $P$ is faithful w.r.t. a DAG $G$ if $X \perp Y | Z \Rightarrow X \perp_G Y | Z$.

123  **Canonical dataset**: We say a discrete dataset $D$ is canonical if its underlying probability distribution
124  $P$ is Markovian and faithful w.r.t. some DAG $G$.

## 3.2 Causal Structure Identifiability

Identifiability discusses which parts of the causal structure can in principle be inferred from the distribution. Below we present the established theory of identifiability on discrete data.

**Causal sufficiency**: There are no latent common causes of any of the variables in the graph.

**Definition 1** (Markov equivalence). *Two graphs are Markov equivalent if and only if they have same skeleton and same v-structures. A Markov equivalence class can be represented by a CPDAG having both directed and undirected edges. A CPDAG can be derived from a DAG $G$, denoted as $CPDAG(G)$.* The theorem of Markov completeness in [24] states that, under causal sufficiency, we can only identify a causal graph up to its Markov equivalence class on canonical data. Therefore, the (non-)identifiable causal relations are the (un)directed edges in the CPDAG. Formally,

**Definition 2** (Identifiability). *Assuming $P$ is Markovian and faithful w.r.t. a DAG $G$ and causal sufficiency, then each (un)directed edge in $CPDAG(G)$ indicates (non-)identifiable causal relation.*

## 3.3 ML4C Related Notations

**Definition 3** (Skeleton). *A skeleton $E$ defined over distribution $P(X_1, X_2, \cdots, X_d)$ is an undirected graph such that there is an edge between $X_i$ and $X_j$ if and only if $X_i$ and $X_j$ are always dependent, i.e., $\nexists Z \subseteq \{X_1, X_2, \cdots, X_d\}$ s.t. $X_i \perp X_j | Z$.* Skeleton is a statistical concept, which can be obtained prior to facilitating various downstream tasks. Recently, there have been some novel skeleton learning algorithms such as [10]. In particular, skeleton can be used for causal learning: theorem in [32] states that if distribution $P$ is Markovian and faithful w.r.t. a DAG $G$, then skeleton $E$ is the same as the undirected graph of $G$.

**Definition 4** (UT). *A triple of variables $\langle X, T, Y \rangle$ in a skeleton is an unshielded triple, or short for UT, if $X$ and $Y$ are adjacent to $T$ but are not adjacent to each other. $\langle X, T, Y \rangle$ can be further oriented to become a v-structure $X \to T \leftarrow Y$, in which $T$ is called the collider.*

**Definition 5** (PC). *Denote the set of parents and children of $X$ in a skeleton as $PC_X$, in other words, $PC_X$ are the neighbors of $X$ in the skeleton. For convenience, if we discuss $PC_X$ in the context of a UT $\langle X, T, Y \rangle$, we intentionally mean the set of parents and children of $X$ but exclude $T$.*

**Definition 6** (Vicinity). *We define the vicinity of a UT $\langle X, T, Y \rangle$ as $V_{\langle X,T,Y \rangle} := \{X, T, Y\} \cup PC_X \cup PC_Y \cup PC_T$. Vicinity is a generalized version of PC, i.e., the neighbors of $\{X, T, Y\}$ in the skeleton.*

**ML4C's training set**: The training set is a collection of discrete datasets $D_1, \cdots, D_n$, where each dataset $D_i$ is associated with a ground truth DAG $G_i$, such that $D_i$ is sampled from $G_i$. $G_i$ derives labels (depends on the chosen learning target), thus $\{D_i, G_i\}_{i \in \{1, \cdots, n\}}$ form ML4C's training set. We can sample graphs from DAG space and generate corresponding datasets, thus obtaining training set in our problem is straightforward.

# 4 Approach

## 4.1 Overview

The core of ML4C is a binary classifier called ML4C-Learner, which takes the orientation of a UT as its learning target, i.e., it classifies whether an input UT $\langle X, T, Y \rangle$ is a v-structure (orientation: $X \to T \leftarrow Y$) or not (orientation remains unknown). Figure 1(b) depicts the overall workflow of ML4C, which is composed of ML4C-Learner with other important inductive biases. Starting from an input dataset $D_i$ with corresponding skeleton $E_i$, we first obtain all the UTs from $E_i$. Featurization is then conducted to represent each UT as an embedded vector, which is further fed into ML4C-Learner. In the inference stage, we obtain all the v-structures which are classified by ML4C-Learner and reconstruct a partial DAG and then, a CPDAG is output by applying Meek rules on the partial DAG. In the training stage, the label of each UT is obtained by querying from ground truth DAG $G_i$. We collect labeled data from multiple datasets in ML4C's training set.

**Proposition 2.** *If ML4C-Learner is a perfect classifier, then ML4C outputs correct CPDAG of a canonical dataset (i.e., ML4C is perfect).*

By Markov completeness, the set of v-structures is invariant across all Markov equivalent DAGs for a canonical dataset, and it can fully recover the CPDAG, provided that the skeleton is given. Thus,

besides its dedicated role in phase 2, ML4C-Learner also facilitates learning task in phase 1 since an identifiable UT implies that it is a v-structure (up to the partial DAG before applying Meek rules).

## 4.2 Featurization

We propose a principled method for ML4C-Learner's featurization, which avoids the need of hand-crafted features. More importantly, we further prove that ML4C-Learner is asymptotically perfect.

**Design Principles**: Our key aspect of featurization is to broaden focus from a specific UT $\langle X, T, Y \rangle$ to its *vicinity* and seeking conditional *dependencies* and structural *entanglement* within the vicinity, to reveal reliable and robust asymmetry to distinguish v-structure and non-v-structure UTs. Specifically, conditional dependencies are the key materials for traditional causal learning methods (e.g., conditional independences for constraint-based methods), and structural entanglement (e.g., $PC_X = PC_T$) are relevant to identifiability: higher entanglement makes the UT less likely to be identifiable.

### • *Dependencies within Vicinity*

**Conditional dependency**: Denoted as $X \sim Y | \mathbf{Z}$ , which is a non-negative scalar that measures the dependence between two random variables $X$ and $Y$ given variable set $\mathbf{Z}$. Operationally, $X \sim Y | \mathbf{Z}$ is composed of two parts, bivariable $X \sim Y$, and conditional $\mathbf{Z}$. We further extend the definition to allow a set of variables in bivariable, and an ensemble (i.e., a set of set) as conditional:

**Extended conditional dependency**: Denoted as $\mathbf{A} \sim \mathbf{B} | \mathcal{Z} := \{ X \sim Y | \mathbf{Z} : X \in \mathbf{A}, Y \in \mathbf{B}, \mathbf{Z} \in \mathcal{Z} \}$, where $\mathbf{A}$ and $\mathbf{B}$ are set of variables, and $\mathcal{Z}$ is an ensemble. Thus, extended conditional dependency is a set of scalars.

Within the vicinity of $\langle X, T, Y \rangle$, we start from measuring dependencies between $\{X, PC_X\}$ and $\{Y, PC_Y\}$ by conditioning on $\{T, PC_T\}$. Intuitively, if $\langle X, T, Y \rangle$ is a v-structure, conditioning on $T$ or $T$'s descendants tends to strengthen the dependency between $PC_X$ and $PC_Y$ since the paths passing $X - T - Y$ are unblocked; otherwise, conditioning on $T$ tends to weaken the dependency between $PC_X$ and $PC_Y$ because $T$ blocks the paths passing $X - T - Y$. Therefore, such conditional dependencies reflect potential asymmetry to distinguish v-structure and non-v-structure. Formally,

**Definition 7** (Domain of bivariable). *Denoted as* $\mathbb{B} := \{X, PC_X\} \times \{Y, PC_Y\} \equiv \{X \sim Y, X \sim PC_Y, PC_X \sim Y, PC_X \sim PC_Y\}$, *here symbol $\times$ is Cartesian product.*

**Definition 8** (Sepsets). *Denoted as* $\mathcal{S} := \{S : X \perp Y | S, \ S \subset PC_X \cup T, \ or \ S \subset PC_Y \cup T\}$. Under faithfulness assumption, sepsets $\mathcal{S}$ is an ensemble where each item is a subset of variables within the vicinity that d-separates $X$ and $Y$.

**Definition 9** (Domain of conditional). *Denoted as* $\mathbb{C} := \{\varnothing, T, \mathcal{PC}_T\} \vee \{\varnothing, S\} \equiv \{\varnothing, T, \mathcal{PC}_T, \mathcal{S}, \mathcal{S} \vee T, \mathcal{S} \vee \mathcal{PC}_T\}$, *where* $\mathcal{PC}_T := \{\{I\} : I \in PC_T\}$ *which is an ensemble version of $PC_T$, and $\mathcal{S} \vee \mathcal{PC}_T := \{S \cup I : S \in \mathcal{S}, I \in \mathcal{PC}_T\}$. Here symbol $\vee$ is element-wise union.*

We exploit the extended conditional dependencies from $\mathbb{B} \times \mathbb{C}$, i.e., we pick a bivariable from $\mathbb{B}$ and a conditional from $\mathbb{C}$, and calculate the extended conditional dependency. There are in total $|\mathbb{B}| \times |\mathbb{C}| = 24$ extended conditional dependencies.

**Lemma 1.** *Sepsets $\mathcal{S}$ of any UT of a canonical dataset is non-empty.* All proofs are available in the supplementary material.

**Remark 1.** *We intend to restrict the sepsets within the vicinity of $\langle X, T, Y \rangle$. Lemma 1 shows the existence of such d-separation sets within vicinity. Furthermore, searching for all d-separation sets is highly time-consuming, thus the computational cost can also be saved drastically.*

### • *Entanglement within Vicinity*

Structural entanglement reflects complex structure within the vicinity of $\langle X, T, Y \rangle$. Variables $X, Y$ and $T$ can mutually share common neighbors, and their neighbors may also overlap with sepsets $\mathcal{S}$. We call such overlaps structural entanglement. Intuitively, stronger entanglement indicates denser structure of vicinity thus making the UT less likely to be identifiable. Therefore, structural entanglement is an important aspect for featurization. Specifically, we exploit the overlap coefficient [33] to measure the entanglement:

**Definition 10** (Overlap coefficient). $\text{OLP}(\mathbf{A}, \mathbf{B}) := |\mathbf{A} \cap \mathbf{B}| / \min(|\mathbf{A}|, |\mathbf{B}|)$, *where $\mathbf{A}$ and $\mathbf{B}$ are two sets of variables.* We extend this formula to support ensemble as input:

224 **(Extended) Overlap coefficient**: $\text{OLP}\left(\mathbf{A}, \mathcal{S}\right) := \sum_{i=1}^{|\mathcal{S}|} \text{OLP}\left(\mathbf{A}, S_i\right)/|\mathcal{S}|$. Naturally, we consider the
225 entanglement in terms of overlap coefficient on each pair of items in domain $\{PC_X, PC_Y, PC_T, \mathcal{S}\}$.
226 Thus, we use 6 scalars to represent the entanglement within the vicinity of a UT.

227 • *Embedding*

228 We aim to represent the dependencies and entanglement by a feature vector with fixed dimensionality,
229 which can be used to train ML4C-Learner. Regarding each extended conditional dependency $\mathbf{A} \sim$
230 $\mathbf{B}|\mathcal{Z} : \mathbf{A} \sim \mathbf{B} \in \mathbb{B}, \mathcal{Z} \in \mathbb{C}$, it consists of a set of scalars with varied set size across UTs, we
231 adopt the kernel mean embedding technique in [30] to represent each $\mathbf{A} \sim \mathbf{B}|\mathcal{Z}$ as a vector with
232 fixed dimensionality. We further modify the embedding algorithm by adding $\min\{\mathbf{A} \sim \mathbf{B}|\mathcal{Z}\}$ and
233 $\max\{\mathbf{A} \sim \mathbf{B}|\mathcal{Z}\}$ as two additional features. We directly use the 6 scalars to represent structural
234 entanglement without further transformation. We concatenate all the embedded vectors to form the
235 final feature vector, as input for ML4C-Learner.

## 4.3 Learnability

237 We have presented ML4C's featurization and started seeing that conditional dependencies and
238 structural entanglement have potential to reveal asymmetry to distinguish v-structure and non-v-
239 structure UTs. Now we provide rigorous analysis to show that, for a canonical dataset with sufficient
240 samples, ML4C-Learner tends to a perfect classifier. To prove this, we first propose a surrogate object
241 called discriminative predicate:

242 **Definition 11** (Discriminative predicate). *A discriminative predicate is a binary predicate function*
243 *with domain as ML4C's feature set.* A discriminative predicate can be viewed as a special classifier
244 with pre-specified form of mechanism (i.e., not learned from data).

245 **Definition 12** (Weak / Strong discriminative predicate). *Whenever a discriminative predicate takes*
246 *the feature vector of a UT as input, a weak discriminative predicate satisfies one of the following*
247 *two criteria; a strong discriminative predicate satisfies both: i) it is evaluated to* TRUE *if the UT is a*
248 *v-structure; ii) it is evaluated to* FALSE *if the UT is not a v-structure.*

249 By definition, a weak discriminative predicate exhibits discriminative power since it is evaluated false
250 implies the UT is a non-v-structure (or true implies v-structure). A strong discriminative predicate
251 can be viewed as a perfect classifier. Denote $\{\mathbf{A} \sim \mathbf{B}|\mathcal{Z}\} > \delta := X \sim Y|\mathbf{Z} > \delta : \forall X \in \mathbf{A}, Y \in$
252 $\mathbf{B}, \mathbf{Z} \in \mathcal{Z}$, then we have:

253 **Lemma 2** (Existence of weak discriminative predicate). *For a canonical dataset with infinite samples,*
254 *the following are three weak discriminative predicates: i)* $\{X \sim Y|T\} > 0$*, ii)* $\{X \sim Y|\mathcal{PC}_T\} = 0$
255 *, iii)* $\{PC_X \sim PC_Y|\mathcal{S} \cup T\} > 0$*.*Take $\{X \sim Y|T\} > 0$ as an example, $\langle X, T, Y \rangle$ is a v-structure
256 $\Rightarrow T$ is a collider $\Rightarrow T$ unblocks $X$ and $Y$ through path $X - T - Y \Rightarrow \{X \sim Y|T\} > 0 \Rightarrow$
257 $\min\{X \sim Y|T\} > 0$, where $\min\{X \sim Y|T\}$ is a feature of ML4C-Learner since $X \sim Y \in$
258 $\mathbb{B}, \{T\} \in \mathbb{C}$.

259 **Lemma 3** (Existence of strong discriminative predicate). *For a canonical dataset with infinite*
260 *samples, the following are three strong discriminative predicates: i)* $\text{OLP}(T, \mathcal{S}) = 0$*, ii)* $\text{OLP}(T, \mathcal{S}) <$
261 $0.5$*, iii)* $\text{OLP}(T, \mathcal{S}) < 1 \land \min\{X \sim Y|T \cup S\} > 0$*.*

262 **CPC/MPC/GLL-MB as special cases of ML4C-Learner**: Predicate $\text{OLP}\left(T, \mathcal{S}\right) = 0 \iff \forall S \in$
263 $\mathcal{S}$, $T \notin S$, which states that the predicate is TRUE if $T$ is not in any d-separation set of $X$ and $Y$.
264 Having correct skeleton provided, this is the criterion of Conservative PC algorithm (CPC) [25] for
265 identifying v-structures. Thus, CPC can be viewed as a special case of ML4C by replacing ML4C-
266 Learner with such a pre-specified logic; $\text{OLP}\left(T, \mathcal{S}\right) < 0.5$ indicates that if more than half of the d-
267 separation sets do not contain $T$, then the UT is oriented as a v-structure, which is called majority rule
268 PC algorithm (MPC) [9]; predicate $\text{OLP}\left(T, \mathcal{S}\right) < 1 \land \min\{X \sim Y|T \cup \mathcal{S}\} > 0 \Rightarrow \exists S \in \mathcal{S}, T \notin S$
269 and $X$ and $Y$ are dependent when conditioning on $T \cup S$, which is used for GLL-MB [2] to more
270 securely identify v-structures. These predicates are with suboptimal performance because only a
271 small portion of features are exploited and the overall loss function of training data is disregarded,
272 thus in practice when an appropriate machine learning model is adopted, ML4C-Learner achieves
273 better performance.

274 **Theorem 1.** *ML4C-Learner tends to a perfect classifier on classifying a canonical dataset with*
275 *sufficient samples.*

## 5   Evaluation

**Benchmark Datasets**   We use discrete datasets sampled by all 24 networks from bnlearn reposi-
tory [28] for evaluation. For each network, we sample 1k, 5k, 10k, 15k, 20k records for use.

**ML4C's Training and Inference**   We generate ML4C's training data synthetically (which is also
used for other SCL competitors). Specifically, 400 unique DAGs are randomly generated by two
models: Erdős-Rényi (ER) model [11] and Scale-Free (SF) model [1], with the number of nodes
ranging from 10 to 1,000. A standard random forward data generation process is applied to obtain
10k observational samples for each graph. We further extract UTs from the 400 DAGs, consisting of
97,010 v-structures (label = 1) and 195,691 non-v-structures (label = 0). We use these instances to
train ML4C-Learner, which is implemented by a XGBoost [5] binary classifier with default hyper-
parameters and we use binary cross-entropy as the loss function. Details on our synthesis procedure,
configurations and implementation of ML4C-Learner are available in the supplementary material.

**Competitors**   We categorize state-of-the-art causal learning algorithms from two aspects, supervised
vs. unsupervised, and can or cannot take skeleton as input. We choose Jarfo [12], D2C [4], RCC [20],
and NCC [21] as SCL competitors. Same as ML4C, all these algorithms can and do require skeleton
as input. All these algorithms use ML4C's training set for training but with different learning target
extracted. Regarding unsupervised algorithms, we choose PC [31], Conservative-PC (CPC) [26],
Majority-rule PC (MPC) [7], GLL-MB (GMB) [2], GES [6], Grow-Shrink (GS) [22], Hill-Climbing
(HC) [16], and Conditional Distribution Similarity (CDS) [12]. which can also take skeleton as input.
Lastly, we also compare with DAG-GNN (DGNN) [35], BLIP [27], and GOBNILP (GNIP) [8],
which are unsupervised algorithms but cannot take skeleton as input.  All these competitors are
capable of dealing with discrete data. All experiments are done in a Windows Server with 2.8GHz
Intel E5-2680 CPU and 256G RAM. Details are in the supplementary material.

**Design**   Our evaluation mainly consists of two parts: end-to-end comparison with competitors on
benchmark datasets, and in-depth experiments on ML4C's learnability. The latter is further divided
into four aspects: i) **Towards a perfect classifier**. As stated in proposition proposition 2, ML4C-
Learner is the core component and we would like to know how far it is from a perfect classifier.
ii) **Reliability** (against weak / strong discriminative predicates). As stated in lemma 2 and 3, there
exist weak and strong discriminative predicates, which have discriminative power and thus are helpful
for ML4C-Learner. Some strong discriminative predicates are equivalent to specific logics of existing
work such as CPC or GLL-MB. Thus, we would like to see how ML4C-Learner takes the advantage
of machine learning, to learn a more reliable classification mechanism (which is also latent and
more sophisticated) than individual weak / strong discriminative predicates. iii) **Robustness** (against
varied sample size). It is known that many causal learning algorithms lack robustness w.r.t sample
noise for finite datasets [20], especially CI tests are error-prone on small samples for constraint-
based algorithms. We would like to evaluate the robustness of ML4C (i.e., the latent classification
mechanism) against varied sample sizes. iv) **Transferability**. It's important for a machine learning
model to generalize well to various types of testing data which are different from training data, such
as different scale (#nodes), graph sparsity, different generating mechanisms, etc.

**Metrics**   We use two standard metrics for performance evaluation: Structural Hamming Distance
(SHD) and F1-score. For each dataset, we measure the SHD / F1-score of the output CPDAG (learned
by a specific algorithm) against the ground truth CPDAG. Specifically, SHD is calculated at CPDAG
level, which is the smallest number of edge additions, deletions, direction reversals and type changes
(directed vs. undirected) to convert the output CPDAG to ground truth CPDAG. F1-score is calculated
over identifiable edges. Roughly, F1-score can be viewed as a normalized version of SHD. Now we
present the experiment results:

**End-to-End Comparison**   Due to page limit, we report SHD and F1-score of all algorithms on
19 large-scale datasets (full results including other 5 smallest and trivial datasets are available in
the supplementary material), as depicted in Table 1. '-' means the algorithm fails on the dataset
(either out-of-memory / exceeds 24 hours execution time / break caused by unknown errors). ML4C
significantly outperforms all other competitors. The average F1-score of ML4C is the highest (0.92,
first column in Table 2). Moreover, ML4C exhibits the most stable performance across all datasets,
its average ranking is $1.5 \pm 0.7$, while the second best is GLL-MB (GMB), with average ranking

Table 1: Experiment results for end-to-end comparison with SOTA causal learning algorithms on benchmark datasets. Algorithm names are abbreviated. SHD and F1-score are reported. The last two rows show statistics of rank by SHD and F1-score for all competitors (Note: F1-score is at UT level).

| Datasets #nodes/#edges | | supervised | | | | | unsupervised | | | | | | | | no skeleton input | | |
|---|---|---|---|---|---|---|---|---|---|---|---|---|---|---|---|---|---|
| | | ML4C | Jarfo | D2C | RCC | NCC | PC | CPC | MPC | GMB | GES | GS | HC | CDS | DGNN | BLIP | GNIP |
| child 20/25 | SHD | **0** | 18 | 16 | 18 | 20 | 22 | 13 | 9 | 20 | 15 | 13 | 13 | 18 | 23 | **0** | **0** |
| | F1 | **1.0** | .24 | .43 | .33 | .12 | .12 | .00 | .74 | .12 | .47 | .59 | .57 | .34 | .25 | **1.0** | **1.0** |
| insurance 27/52 | SHD | **5** | 41 | 30 | 34 | 28 | 36 | 34 | 21 | 29 | 34 | 28 | 19 | 36 | 53 | 35 | 14 |
| | F1 | **.89** | .26 | .44 | .42 | .44 | .39 | .00 | .66 | .55 | .46 | .56 | .76 | .36 | .05 | .51 | .82 |
| water 32/66 | SHD | 5 | 33 | 43 | 31 | **0** | 4 | 60 | 7 | 8 | 38 | 27 | 38 | 18 | 61 | 65 | 52 |
| | F1 | .94 | .52 | .34 | .56 | **1.0** | .97 | .00 | .91 | .87 | .49 | .62 | .46 | .76 | .00 | .20 | .50 |
| mildew 35/46 | SHD | 6 | - | 17 | 25 | 34 | 21 | - | - | 7 | **3** | 9 | 23 | 18 | 52 | 36 | - |
| | F1 | .87 | - | .68 | .50 | .33 | .56 | - | - | .85 | **.93** | .80 | .64 | .65 | .19 | .41 | - |
| alarm 37/46 | SHD | **1** | 21 | 26 | 18 | 20 | 20 | 20 | 6 | 17 | 8 | 3 | 21 | 18 | 46 | 17 | 2 |
| | F1 | **.98** | .57 | .44 | .64 | .57 | .57 | .57 | .92 | .64 | .86 | .94 | .66 | .62 | .12 | .82 | .98 |
| barley 48/84 | SHD | 5 | 48 | 55 | 50 | **0** | 3 | - | - | 8 | 42 | - | 34 | 50 | 87 | 60 | 42 |
| | F1 | .95 | .46 | .38 | .44 | **1.0** | .96 | - | - | .91 | .59 | - | .72 | .43 | .00 | .48 | .67 |
| hailfinder 56/66 | SHD | 11 | 47 | 41 | 43 | **0** | 17 | - | - | 26 | 60 | - | 59 | 44 | 76 | 111 | 118 |
| | F1 | .80 | .37 | .45 | .42 | **1.0** | .85 | - | - | .70 | .21 | - | .23 | .42 | .00 | .18 | .12 |
| hepar2 70/123 | SHD | **0** | 54 | 81 | 59 | **0** | 35 | 27 | 37 | 14 | 46 | 40 | 35 | 75 | 123 | 79 | 61 |
| | F1 | **1.0** | .59 | .34 | .54 | **1.0** | .72 | .81 | .70 | .89 | .75 | .70 | .81 | .39 | .00 | .54 | .68 |
| win95pts 76/112 | SHD | 1 | 65 | 51 | 33 | **0** | 8 | 42 | 7 | 5 | 32 | 21 | 16 | 50 | 112 | 103 | - |
| | F1 | .99 | .43 | .54 | .73 | **1.0** | .95 | .64 | .95 | .97 | .77 | .85 | .91 | .57 | .00 | .47 | - |
| pathfinder 109/195 | SHD | 25 | 157 | 145 | 151 | **0** | 150 | - | - | 147 | 158 | - | 168 | 148 | 196 | 241 | - |
| | F1 | .77 | .21 | .29 | .21 | **1.0** | .29 | - | - | .30 | .29 | - | .28 | .31 | .00 | .07 | - |
| munin1 186/273 | SHD | **10** | 169 | 154 | 153 | 72 | 86 | 117 | - | 84 | 109 | - | 233 | 151 | - | 257 | - |
| | F1 | **.97** | .42 | .47 | .46 | .77 | .71 | .58 | - | .72 | .67 | - | .26 | .50 | - | .42 | - |
| andes 223/338 | SHD | **0** | 226 | 209 | 246 | **0** | 4 | 83 | 4 | 5 | 47 | 15 | 38 | 149 | - | 175 | - |
| | F1 | **1.0** | .35 | .41 | .29 | **1.0** | .99 | .75 | .99 | .98 | .92 | .96 | .92 | .60 | - | .76 | - |
| diabetes 413/602 | SHD | 25 | 220 | 395 | 237 | 48 | **0** | - | - | 204 | 146 | - | 592 | 368 | - | 534 | - |
| | F1 | .96 | .62 | .38 | .62 | .96 | **1.0** | - | - | .68 | .77 | - | .03 | .43 | - | .43 | - |
| pigs 441/592 | SHD | **0** | 350 | 332 | 263 | 400 | 400 | - | - | 268 | **0** | - | 532 | 316 | - | 6 | - |
| | F1 | **1.0** | .44 | .46 | .59 | .35 | .35 | - | - | .56 | **1.0** | - | .18 | .50 | - | 1.0 | - |
| link 724/1125 | SHD | **0** | 731 | 630 | 638 | 749 | 737 | - | - | 204 | 324 | - | 1047 | 400 | - | 947 | - |
| | F1 | **1.0** | .38 | .45 | .45 | .39 | .40 | - | - | .81 | .80 | - | .14 | .64 | - | .49 | - |
| munin 1041/1397 | SHD | 72 | 967 | 790 | 816 | **0** | 156 | - | - | 458 | 661 | - | 1397 | 795 | - | 1599 | - |
| | F1 | .95 | .36 | .48 | .44 | **1.0** | .89 | - | - | .69 | .62 | - | .00 | .51 | - | .29 | - |
| munin2 1003/1244 | SHD | **118** | 554 | 611 | 646 | 1052 | 898 | - | - | 536 | 632 | - | 1240 | 753 | - | 1321 | - |
| | F1 | **.92** | .60 | .56 | .55 | .19 | .30 | - | - | .57 | .58 | - | .01 | .49 | - | .46 | - |
| munin3 1041/1306 | SHD | **113** | 616 | 629 | 688 | 1048 | 860 | - | - | 544 | 566 | - | 1306 | 819 | - | 1539 | - |
| | F1 | **.92** | .58 | .57 | .54 | .25 | .37 | - | - | .60 | .65 | - | .00 | .46 | - | .26 | - |
| munin4 1038/1388 | SHD | **126** | 696 | 658 | 776 | 1058 | 876 | - | - | 649 | 618 | - | 1388 | 812 | - | 1627 | - |
| | F1 | **.93** | .54 | .56 | .50 | .29 | .39 | - | - | .55 | .64 | - | .00 | .49 | - | .28 | - |
| rank(SHD) | mean ±stdd | **1.5** **0.7** | 9.1 3.2 | 8.2 3.7 | 7.9 2.2 | 5.1 4.2 | 6.3 3.7 | 10.8 2.9 | 9.5 4.1 | 4.4 2.4 | 5.8 2.9 | 9.6 3.6 | 8.7 2.7 | 7.9 2.4 | 13.3 1.8 | 10.5 3.5 | 10.7 4.3 |
| UT-F1 | mean ±stdd | **.90** **.13** | .22 .17 | .19 .13 | .27 .18 | .66 .40 | .50 .34 | .53 .33 | .87 .16 | .59 .32 | .54 .28 | .77 .24 | .47 .35 | .30 .22 | .09 .07 | .36 .29 | .70 .33 |

Table 2: Reliability: average F1-score of ML4C vs. 8 discriminative predicates extracted from ML4C features on benchmark datasets.

| | | strong predicates | | | | weak predicates | | | |
|---|---|---|---|---|---|---|---|---|---|
| id | ML4C | 1 | 2 | 3 | 4 | 1 | 2 | 3 | 4 |
| F1 | **.92**±**.20** | .77±.31 | .52±.27 | .38±.25 | .66±.27 | .72±.25 | .61±.29 | .73±.30 | .55±.27 |

$4.4 \pm 2.4$. Among the competitors, NCC ranks #1 on 8 datasets (note that ML4C ranks #1 on 11 datasets), but its performance fluctuates. Overall it only ranks $5.1 \pm 4.2$. Last but not least, ML4C shows high accuracy (F1>0.9) on very large-scale datasets (e.g., medicine datasets 'munin*' [3]) while $\max(others) \sim 0.6$.

**Towards a Perfect Classifier** The last row of Table 1 shows the performance of ML4C-Learner component at UT level by UT-F1 (i.e., F1-score of classifying UTs): such UT level accuracy is crucial for causal learning on discrete data, since the set of v-structures is invariant across all Markov equivalent DAGs and it can fully recover the CPDAG. The average F1-score of ML4C-Learner is $0.90 \pm 0.13$, which shows promising results towards a perfect classifier.

Table 3: Robustness: ML4C is trained on synthetic datasets with sample size = 10k, but tested on benchmark datasets with different sample sizes $\in$ {1k, 5k, 10k, 15k, 20k}.

| | size | 1k | 5k | 10k | 15k | 20k | size | 1k | 5k | 10k | 15k | 20k | size | 1k | 5k | 10k | 15k | 20k |
|---|---|---|---|---|---|---|---|---|---|---|---|---|---|---|---|---|---|---|
| SHD | insurance | 11 | 1 | 5 | 1 | 0 | water | 12 | 11 | 5 | 8 | 6 | mildew | 8 | 5 | 3 | 6 | 1 |
| F1 | 27/52 | .81 | .97 | .89 | .97 | 1.0 | 32/66 | .86 | .87 | .94 | .89 | .93 | 35/46 | .83 | .89 | .93 | .87 | .98 |
| SHD | alarm | 5 | 4 | 0 | 1 | 5 | barley | 13 | 9 | 4 | 8 | 6 | hailfinder | 15 | 15 | 6 | 15 | 13 |
| F1 | 37/46 | .93 | .95 | 1.0 | .98 | .93 | 48/84 | .88 | .93 | .97 | .92 | .94 | 56/66 | .74 | .72 | .90 | .72 | .76 |
| SHD | hepar2 | 8 | 2 | 0 | 1 | 2 | win95pts | 7 | 1 | 0 | 1 | 1 | pathfinder | 1 | 7 | 25 | 7 | 1 |
| F1 | 70/123 | .96 | .99 | 1.0 | .99 | .99 | 76/112 | .96 | .99 | 1.0 | .99 | .99 | 109/195 | .99 | .92 | .77 | .92 | .99 |
| SHD | munin1 | 32 | 7 | 10 | 9 | 15 | andes | 3 | 2 | 0 | 2 | 0 | diabetes | 18 | 28 | 4 | 26 | 27 |
| F1 | 186/273 | .89 | .98 | .97 | .97 | .95 | 223/338 | .99 | .99 | 1.0 | .99 | 1.0 | 413/602 | .97 | .95 | .99 | .96 | .96 |
| SHD | pigs | 0 | 0 | 0 | 0 | 0 | link | 88 | 13 | 0 | 0 | 0 | munin | 107 | 76 | 71 | 93 | 87 |
| F1 | 441/592 | 1.0 | 1.0 | 1.0 | 1.0 | 1.0 | 724/1125 | .93 | .99 | 1.0 | 1.0 | 1.0 | 1041/1397 | .93 | .95 | .96 | .94 | .94 |
| SHD | munin2 | 117 | 95 | 120 | 110 | 97 | munin3 | 151 | 119 | 113 | 99 | 62 | munin4 | 165 | 130 | 123 | 146 | 133 |
| F1 | 1003/1244 | .92 | .93 | .92 | .93 | .93 | 1041/1306 | .90 | .92 | .92 | .94 | .96 | 1038/1388 | .90 | .92 | .93 | .91 | .93 |

Table 4: Transferability: ML4C trains/tests both on synthetic datasets with different configurations.

| | train | test | SHD | F1 | test | SHD | F1 | test | SHD | F1 | test | SHD | F1 |
|---|---|---|---|---|---|---|---|---|---|---|---|---|---|
| # node | 10 | 10 | 1.2±2.4 | .94±.12 | 50 | 4.8±3.4 | .95±.03 | 100 | 6.6±4.7 | .97±.02 | 1k | 50.6±8.4 | .97±.00 |
| | 50 | 10 | 0.4±0.8 | .97±.05 | 50 | 0.8±1.0 | .99±.01 | 100 | 4.4±4.7 | .98±.02 | 1k | 23.2±5.7 | .99±.00 |
| | 100 | 10 | 0.0±0.0 | 1.0±.00 | 50 | 1.2±1.6 | .99±.01 | 100 | 4.0±4.6 | .98±.02 | 1k | 21.6±4.8 | .99±.00 |
| | 1k | 10 | 0.4±0.8 | .97±.05 | 50 | 0.8±1.0 | .99±.01 | 100 | 1.4±2.3 | .99±.01 | 1k | 14.8±8.2 | .99±.00 |
| sparsity | 1 | 1 | 0.8±1.6 | .99±.02 | 2 | 3.4±2.9 | .97±.02 | 3 | 3.0±2.5 | .98±.01 | 4 | 11.4±3.9 | .95±.02 |
| | 2 | 1 | 1.8±1.6 | .98±.02 | 2 | 2.2±1.7 | .98±.01 | 3 | 2.2±2.0 | .99±.01 | 4 | 8.2±2.5 | .97±.01 |
| | 3 | 1 | 1.0±1.3 | .98±.02 | 2 | 2.2±1.3 | .98±.01 | 3 | 4.4±3.6 | .97±.02 | 4 | 4.0±3.2 | .98±.01 |
| | 4 | 1 | 2.4±2.3 | .97±.03 | 2 | 2.2±1.9 | .98±.01 | 3 | 3.2±2.7 | .98±.02 | 4 | 4.8±3.7 | .98±.01 |
| samplesize | 1k | 1k | 2.8±2.3 | .97±.02 | 5k | 2.0±2.2 | .98±.02 | 10k | 1.6±2.3 | .98±.02 | 20k | 1.0±1.3 | .99±.01 |
| | 5k | 1k | 5.2±2.9 | .95±.03 | 5k | 1.0±2.0 | .99±.02 | 10k | 2.2±3.5 | .98±.04 | 20k | 0.6±0.8 | .99±.01 |
| | 10k | 1k | 5.2±4.8 | .95±.05 | 5k | 1.8±2.7 | .98±.02 | 10k | 2.0±3.1 | .98±.03 | 20k | 0.6±0.8 | .99±.01 |
| | 20k | 1k | 4.8±3.3 | .95±.03 | 5k | 2.4±2.6 | .98±.02 | 10k | 1.2±1.6 | .99±.02 | 20k | 1.0±1.3 | .99±.02 |
| gtype | ER | ER | 1.0±2.0 | .99±.02 | SF | 2.2±1.6 | .98±.01 | | | | | | |
| | SF | ER | 1.6±1.9 | .98±.02 | SF | 2.2±2.4 | .98±.02 | | | | | | |

**Reliability** We manually identify 4 strong discriminative predicates and 4 weak discriminative predicates and treat each one as a replacement of ML4C-Learner. Table 2 shows the performance of these predicates. Although most predicates show value on discriminating UTs (e.g., 5/8 predicates are with >0.6 F1-score), ML4C-Learner has higher performance (average F1-score = 0.92) than each individual predicate (best average F1-score = 0.77). Thus, it is evident that ML4C-Learner learns a more reliable classification mechanism, by taking advantage of machine learning techniques.

**Robustness** To evaluate robustness, ML4C is trained on synthetic datasets with sample size = 10k, but it is tested on benchmark datasets with different sample sizes: 1k, 5k, 10k, 15k and 20k respectively. Table 3 shows that ML4C exhibits satisfactory robustness (decrease of F1-score is less than 0.1) against sample size on most datasets (17/18, except for 'hailfinder').

**Transferability** To evaluate whether ML4C generalizes well to various types of testing data, we vary scale (#nodes), graph sparsity, generating mechanism and sample size. ML4C is trained on a fixed configuration but it is tested with different domains (i.e., data generated under different configuration). Result is depicted in Table 4, ML4C transfers well on different domains, e.g., even if it is trained on 10 nodes but tested on 1,000 nodes (last column of the first row in Table 4), the F1-score only drops 0.02.

## 6 Conclusion and Future Work

We have proposed a supervised causal learning algorithm ML4C, with theoretical guarantee on learnability and remarkable empirical performance. More importantly, ML4C shows promising results on validating the effectiveness of supervision. To make SCL practical in real-world scenarios, one important direction for future work is to identify reliable and accurate skeleton from data, considering ML4C requires skeleton as additional input.

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
