# A  Proofs of Lemmas and Theorems

## A.1  Proof of Proposition 1

**Proposition 1.** *If the learning target is non-identifiable (i.e., every edge in the target is non-identifiable) a priori, then SCL is not better than random guessing.*

**Re-statement:** We take learning target as the orientation of an edge as an example, so we are analyzing the performance of a binary classifier against random guessing. The conclusion can be easily extended to general case.

Denote random guessing as a degenerated estimator $r(X) \equiv 0.5$, which indicates the probability of label = 1 is always 0.5, regardless of any input.

Denote the joint probability distribution of $X$ and $Y$ as $\mathbb{P}$ and the space of all joint probability distribution is $\mathcal{P}$, then we aim to prove the following statement which is in an adversarial setting:

$$r = \arg\min_{f \in \mathcal{C}} \sup_{\mathbb{P} \in \mathcal{P}} E_{(X,Y) \sim \mathbb{P}} \left[ -Y \log f(X) - (1 - Y) \log (1 - f(X)) \right]$$

The expectation is the standard binary cross entropy loss; we are allowed to enumerate every possible joint probability distribution in $\mathcal{P}$ because the learning target is non-identifiable. $\mathcal{C}$ is the space of all possible binary classifiers.

*Proof.* Given any binary classifier $f$, we partition the space of $X$ by $A$, $B$ and $C$ where $A = \{x | f(x) > 0.5\}$, $B = \{x | f(x) < 0.5\}$, $C = \{x | f(x) = 0.5\}$. Then we construct the following joint probability distribution $P^*$:

$$P^*(X,Y) = \left\{ \begin{array}{l} P^*(Y = 0 | X = x) = 1 \text{ if } x \in A \\ P^*(Y = 1 | X = x) = 1 \text{ if } x \in B \\ \text{arbitrary if } x \in C \end{array} \right\}$$

Then it is easy to see that $E_{(X,Y) \sim P^*} \left[ -Y \log f(X) - (1 - Y) \log (1 - f(X)) \right] \geq 1$. Note that $E \left[ -Y \log r(X) - (1 - Y) \log (1 - r(X)) \right] \equiv 1$, thus $r$ achieves minimum worse-case loss. $\square$

## A.2  Proof of Proposition 2

**Proposition 2.** *If ML4C-Learner is a perfect classifier, then ML4C outputs correct CPDAG of a canonical dataset (i.e., ML4C is perfect).*

*Proof.* Classical constraint-based methods consist of three steps: skeleton identification, v-structure identification, and further edge orientation by applying Meek rules [37]. It has been proved in PC [34] that when learning from a canonical dataset, if both the identified skeleton and v-structures are correct, then the learned CPDAG is correct. ML4C follows the three steps, with the correct skeleton is given as input, and ML4C-Learner is responsible for v-structure identification. Thus, assuming ML4C-Learner is a perfect classifier (i.e., correctly identifies all v-structures) implies that ML4C outputs correct CPDAG. $\square$

## A.3  Proof of Lemma 1

**Lemma 1.** *Sepsets $\mathcal{S}$ of any UT of a canonical dataset is non-empty.*

*Proof.* According to Lemma 3.3.9 of [35], in a directed acyclic graph $G$, if $X$ is not a descendant of $Y$, and $X$ and $Y$ are not adjacent, then $X$ and $Y$ are d-separated by **Parents**$(Y)$. Given an UT $\langle X, T, Y \rangle$, $X$ and $Y$ are not adjacent. Either $X$ is not a descendant of $Y$, or $Y$ is not a descendant of $X$, otherwise a loop will be introduced. Thus there either exists **Parents**$(X) \equiv PC_X \cup T$, or **Parents**$(Y) \equiv PC_Y \cup T$, which belongs to $\mathcal{S}$. Thus $\mathcal{S}$ is non-empty. $\square$

## A.4  Proof of Lemma 2

**Lemma 2** (Existence of weak discriminative predicate)**.** *For a canonical dataset with infinite samples, the following are three weak discriminative predicates: i) $\{X \sim Y | T\} > 0$, ii) $\{X \sim Y | \mathcal{PC}_T\} = 0$, iii) $\{PC_X \sim PC_Y | \mathcal{S} \cup T\} > 0$.*

*Proof.* For a canonical dataset with infinite samples,

1. $\{X \sim Y | T\} > 0$: 1) $\langle X, T, Y \rangle$ is a v-structure $\Rightarrow T$ is a collider $\Rightarrow T$ unblocks $X$ and $Y$ through path $X - T - Y \Rightarrow \{X \sim Y | T\} > 0$ holds TRUE. 2) if $\langle X, T, Y \rangle$ is not a v-structure, then $\{X \sim Y | T\} > 0$ can be TRUE or FALSE. e.g., it is FALSE for $X \to T \to Y$ (no more paths connect $X$ and $Y$), but if there exists another node $X \to T' \to Y$, it is evaluated TRUE. Therefore, it satisfies criterion ii) of definition 12, but not i) hence it is a weak discriminative predicate.

2. $\{X \sim Y | \mathcal{PC}_T\} = 0$: 1) $\langle X, T, Y \rangle$ is not a v-structure $\Rightarrow T$ is a non-collider $\Rightarrow \forall pc_t \in \mathcal{PC}_T$, there exists a path $X - T - Y$ from $X$ to $Y$, where $T$ is the only node on path, $T$ is a non-collider, and $T \notin \{pc_t\} \Rightarrow pc_t$ does not block the path $\Rightarrow \{X \sim Y | \mathcal{PC}_T\} = 0$ always holds FALSE. 2) if $\langle X, T, Y \rangle$ is a v-structure, then $\{X \sim Y | \mathcal{PC}_T\} = 0$ can be TRUE or FALSE. Therefore, it satisfies criterion i) but not ii) hence it's a weak discriminative predicate.

3. $\{PC_X \sim PC_Y | \mathcal{S} \cup T\} > 0$: 1) $\langle X, T, Y \rangle$ is a v-structure $\Rightarrow T$ is a collider $\Rightarrow \forall pc_x \in PC_X, pc_y \in PC_Y, S \in \mathcal{S}, S \cup T$ unblock $pc_x$ and $pc_y$ through path $pc_x - X - T - Y - pc_y \Rightarrow \{pc_x \sim pc_y | S \cup T\} > 0$ always hold TRUE. 2) if $\langle X, T, Y \rangle$ is not a v-structure then it can be TRUE or FALSE. Therefore, it satisfies criterion ii) but not i) hence it's a weak discriminative predicate.

$\square$

## A.5 Proof of Lemma 3

**Lemma 3** (Existence of strong discriminative predicate). *For a canonical dataset with infinite samples, the following are three strong discriminative predicates: i)* OLP$(T, \mathcal{S}) = 0$, *ii)* OLP$(T, \mathcal{S}) < 0.5$, *iii)* OLP$(T, \mathcal{S}) < 1 \wedge \min \{X \sim Y | T \cup S\} > 0$.

*Proof.* First, it is known that the following three algorithms are sound and complete for a canonical dataset with infinite samples: CPC [28], MPC [8] and GLL-MB [3]. Below we translate each predicate and then show that these predicates are equivalent to the criterion to identify v-structures in CPC [28], MPC [8] and GLL-MB [3] respectively.

1. Predicate OLP$(T, \mathcal{S}) = 0 \iff \forall S \in \mathcal{S}, T \notin S$, which states that predicate is TRUE if and only if $T$ is not in any d-separation set of $X$ and $Y$. This is exactly the criterion of CPC for identifying v-structures [28].

2. Predicate OLP$(T, \mathcal{S}) < 0.5$ indicates that only if more than half of the d-separation sets do not contain $T$, then the UT is oriented as a v-structure. This is called majority-rule PC algorithm MPC [8] for v-structure identification.

3. Predicate OLP$(T, \mathcal{S}) < 1 \wedge \min \{X \sim Y | T \cup \mathcal{S}\} > 0 \Rightarrow \exists S \in \mathcal{S}, T \notin S$ and $X$ and $Y$ are dependent when conditioning on $T \cup S$, which is the criterion used for GLL-MB [3] to identify v-structures.

$\square$

## A.6 Proof of Theorem 1

**Theorem 1.** *ML4C-Learner tends to a perfect classifier on classifying a canonical dataset with sufficient samples.*

*Proof.* According to Lemma 3, there exists strong discriminative predicate $P$ which achieves zero loss given a canonical dataset and sufficient samples. Thus, when adequate ML model is chosen, ML4C-Learner can achieve no worse performance than $P$ (e.g., we can set the parameters of ML4C-Learner so that it approximates predicate $P$ initially, and then apply standard gradient descent procedure). By considering proposition 2, we complete the proof. $\square$

# B Implementation Details

## B.1 Calculating conditional dependencies

There are several ways to measure the conditional dependence, such as p-value by testing of conditional independence, or conditional mutual information [9]. For categorical variables, a good choice is $G^2$ test [1]. In our implementation, we adopt an approximate version of $G^2$ statistic, and use p-value to measure the conditional dependence.

Moreover, considering p-value can easily vanish due to numerical precision in 64-bit computers. Therefore, we use a transformation of p-value to avoid the issue, as additional quantity to measure conditional dependency. We first define complementary error function as

$$g(z) = 1 - \frac{2}{\sqrt{\pi}} \int_0^z e^{-t^2} \mathrm{d}t,$$

and we use quantity $z$ by inverse of $g$:

$$z = g^{-1}(x).$$

568 Given a p-value $x$, we use $g^{-1}$ as a non-linear transformation to obtain a better re-scaled quantity to measure
569 conditional dependency. Intuitively, $z$ can be viewed as $z$-sigma for a standard normal distribution, e.g., if
570 p-value is 0.05, then $z = 2$, since 2-sigma indicates probability of values that lie within 2-sigma interval in a
571 normal distribution is 0.95.

## B.2 ML4C Training and inference details

### B.2.1 Data synthesis details

574 **Graph structure:** We adopt the Erdős-Rényi (ER) model [13] and the Scale-Free (SF) model [2], which are
575 two commonly used model for graph synthesis. We categorize the scale of the graph (number of nodes $d$) into
576 four classes: small, medium, large, and very large, corresponding to $d$ being uniformly sampled from intervals
577 $[10, 20]$, $[21, 50]$, $[51, 100]$, and $[101, 1000]$, respectively. Given the number of nodes $d$, the sparsity of the
578 graph (defined as the ratio of the average number of edges to the number of nodes, i.e., the average in-degree
579 of all nodes) is randomly sampled from a uniform distribution $[1.2, 1.7]$. Given the number of nodes and the
580 expected number of edges, the graph skeleton is generated accordingly by the two random graph models. Then
581 the skeleton is randomly oriented to a DAG by upper triangular permutation.

582 **Conditional probability table:** Now we illustrate how we come up with Conditional Probability Table (CPT)
583 for each node. In accordance with the topological ordering of the graph, each node is first assigned its cardinality,
584 which is randomly sampled from a truncated normal distribution $\mathcal{N}(\mu = 2, \sigma = \frac{1.5}{m}, \min = 2)$, where $m$
585 denotes the maximum number of peers of the node (i.e. $\max\{\text{in-degree of the effect nodes of this node}\}$). This
586 regularization is designed to make the forward sampling process faster and prevent some certain nodes with
587 many cause nodes from getting stuck. Since the number of different conditions to be enumerated is exponential
588 ($\Pi_{c \in \text{causes}} \text{cardinality}_c$), node with a larger maximum peers number tends to have smaller cardinality. Next, we
589 enumerate each of its unique conditions (given by combinations of its cause nodes' cardinalities) and randomly
590 generate its probability distribution at each condition. The probability distribution is sampled from a Dirichlet
591 distribution with parameter $\alpha \sim U[0.1, 1.0]$ and grid number as this code's cardinality.

592 **Training data:** Having CPT specification of each node, a sample of 10k rows of observations is obtained for
593 each graph according to the standard Bayesian network forward sampling. This generates a total dataset of 4
594 scales $\times$ 2 graph models $\times$ 50 graphs for each class = 400 unique graphs and the corresponding sampled data.
595 Different SCL algorithms are then further used to extract the required features corresponding to the respective
596 learning targets, e.g., all edges of all graphs for pairwise SCL algorithms. For our ML4C learning targets, all UTs
597 are extracted from graphs, consisting of a total of 97,010 V-structures (label=1) and 195,691 non-V-structures
598 (label=0).

### B.2.2 XGBoost hyper-parameter settings

600 We use `xgb.XGBClassifier()`, the Python API provided by XGBoost [6], to implement the binary classifier
601 ML4C-Learner. All hyper-parameters are set as default. We set the threshold value $T = 0.1$.

## B.3 Post processing

603 Although ML4C-Learner achieves high accuracy on classifying UTs into v-structures or non-v-structures (UT-F1
604 = 0.9, as shown in Table 1), it is still possible to have conflicts among the detected v-structures. We adopt
605 a straightforward heuristic to resolve conflicts: suppose we have two conflict v-structures $A \rightarrow B \leftarrow C$
606 and $B \rightarrow C \leftarrow D$, we discard the one with lower probability score (by ML4C-Learner). We continue such
607 pairwise conflict resolving until no more conflicts exist. We use the left v-structures to construct the partial DAG
608 (bottom-right of Figure 1(b)). Pseudo-code is shown in Algorithm 1.

# C  Details of Evaluation

## C.1 Evaluation metrics

611 We calculate SHD at CPDAG level. Specifically, SHD is computed between the learned $\text{CPDAG}(\hat{G})$ and ground
612 truth $\text{CPDAG}(G)$, i.e., the smallest number of edge additions, deletions, direction reversals and type changes
613 (directed vs. undirected) to convert the output CPDAG to ground truth CPDAG. As is shown in Table 5, SHD is
614 equal to the sum of the number of ✗s in the table.

F1-score is then calculated based on the identifiable edges of $\text{CPDAG}(\hat{G})$ and $\text{CPDAG}(G)$, where the accuracy
(precision) is equal to True Positive Rate (TPR) and the recall (recall) is equal to 1 - False Discovery Rate (FDR).

```
input  : v-structure candidates $VC = \{v_1, \cdots, v_p\}$,
            score querier $s : v_i \to s_i$, returning $v_i$'s probability score
output : Final v-structure candidates $FV$, which is self-consistent.
Initialize: removing v-structure set $RV$.
for  $v_i \in VC$ do
    $s_i \leftarrow s(v_i)$
    flag← FALSE
    for  $v_j \in VC$ do
        $s_j \leftarrow s(v_j)$
        if $v_i$ conflicts with $v_j$ and $s_i < s_j$ then
            flag← TRUE
            break
    if flag then
        SV ← SV∪$\{v_i\}$
FV←VC\RV.
```

**Algorithm 1:** Conflict resolving

Table 5: SHD calculation details.

| in result CPDAG→ | iden (directed) | | uniden | missing in |
| in truth CPDAG↓ | right | wrong | (undirected) | skeleton |
| --- | --- | --- | --- | --- |
| iden | ✓ ① | ✗ ② | ✗ ③ | ✗ ④ |
| uniden | ✗ ⑤ | | ✓ ⑥ | ✗ ⑦ |
| nonexist | ✗ ⑧ | | ✗ ⑨ | ✓ ⑩ |

Details about the specific calculation can also refer to Table 5:

$$\text{precision=TPR} = \frac{①}{① + ② + ③ + ④},$$

$$\text{recall=1-FDR} = \frac{①}{① + ② + ⑤ + ⑧},$$

## C.2  Full result of Table 1: End-to-end comparison

Here we report full results including other 5 smallest and trivial datasets. Note that 1) All F1-score degrade into 0. on sachs dataset, because that sachs has no identifiable edges. 2) The rank(SHD) row is also re-calculated over full datasets.

## C.3  Predicates in Table 2: Reliability

Table 2 shows the performance of 4 weak discriminative predicates and 4 strong discriminative predicates. Specifically, the four strong predicates are respectively 1) $t \sim U[0, 1]$, $\text{OLP}(T, \mathcal{S}) \geq t$; 2) $\text{OLP}(T, \mathcal{S}) == 0$; 3) $\text{OLP}(T, \mathcal{S}) == 0$ and $\{X \sim Y | \mathcal{S} \cup T\} > 0$; 4) $\{X \sim Y | \mathcal{S} \vee T\} > 0$. The four weak predicates are respectively 1) $\{PC_X \sim PC_Y | T\} > 0$; 2) $\{PC_X \sim PC_Y | \mathcal{S} \vee T\} > 0$; 3) $\{X \sim Y | PC_T\} == 0$; 4) $\{X \sim Y | \mathcal{S} \vee \mathcal{PC}_T\} > 0$.

## C.4  Details of Table 4: Transferability

To evaluate ML4C's transferability across different domains, we train on dataset generated using one configuration, and test on another. By default the configuration is that: #nodes=50, sparsity=#edges/#nodes=1.5, generating model=ER, and sample size=10000. We conduct controlled trials on the four configuration domains listed above (shown as the four big bars of Table 4).

When we test transferability over one domain (e.g., the first bar, #nodes), then #nodes is set from 4 options (10, 50, 100, 1k), and $4 \times 4 = 16$ pairs of train-test experiments are conducted. For each experiment, 50 graphs are synthesized for training and another 5 graphs for test. Except for the target domain (#nodes), all the other

Table 6: Full result of Table 1.

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

domains use the default configuration. The result SHD and F1-score are reported as mean value and standard deviation over the five test graphs.

# D  Code and Data

## D.1  URLs of all competitors

We use open-source codes of other algorithms for evaluation.

For Jarfo, RCC, NCC, GES, GS(Grow-Shrink), and CDS, we use the API provided by Causal Discovery Toolbox [18]: `https://github.com/FenTechSolutions/CausalDiscoveryToolbox`.

For HC(Hill-Climbing) we use pgmpy `https://github.com/pgmpy/pgmpy` with BDeu score.

For PC we use the official R package pcalg `https://cran.r-project.org/web/packages/pcalg`.

For Conservative-PC and Majority-rule PC, we slightly modify the source code of pcalg to enable a faster run on large scale datasets. GLL-MB is also implemented based on pcalg. Reviewers can download our modified implementation of these 3 algorithms from `http://ml4c.xyz`.

## D.2 Algorithms starting from data: DAG-GNN/BLIP/GOBNILP

### D.2.1 Code URL

1. GOBNILP: `https://bitbucket.org/jamescussens/pygobnilp/`.

2. BLIP: `https://cran.r-project.org/web/packages/r.blip/`.

3. DAG-GNN: We use a repository with a standard and clean version of the DAG-GNN algorithm, which is well maintained and can be found at `https://github.com/ronikobrosly/DAG_from_GNN/`.

### D.2.2 Hyper-parameter settings

1. Time limit: The running time of all programs is limited to 24 hours.

2. Max-in-degree: The max-in-degree threshold for BILP is set to 6. The max-in-degree threshold for GOBNILP is set to 3.

3. Configurations of DAG-GNN are as follows. Epochs=300, batch size=100, learning rate=3e-3, graph threshold=0.3. Graph threshold is a threshold for weighted adjacency matrix (i.e., any weights > -0.3 and < 0.3 means the two variables are not adjacent).

### D.2.3 Verifying the results of DAG-GNN

To make sure DAG-GNN is correctly executed, we have carefully experiment DAG-GNN from the following two aspects:

**Reproducing the results of paper [38]**   We take the child dataset as an example to test the reproducibility, because the data set has been reported by [38]. As can be seen from Table 7, the BIC scores are similar to

Table 7: Reproducing results for child

|  | groundtruth | child |
|---|---|---|
| BIC | -1.23e+4 | -1.36e+4 |

the results reported in the original paper (child: -1.38e+4). That is to say, the results in Yu et al.'s paper are reproduced by us.

**Different graph thresholds**   The following are the BIC scores on the data sets of alarm and water with different graph thresholds. The graph threshold recommended by [38] is 0.3. It can be seen that the performance

Table 8: Results with different graph thresholds

|  | Groundtruth | 0.1 | 0.2 | 0.3 | 0.4 | 0.5 |
|---|---|---|---|---|---|---|
| alarm | -1.08e+5 | -1.90e+5 | -1.44e+5 | -1.59e+5 | -1.77e+5 | -1.91e+5 |
| water | -1.35e+5 | -1.32e+5 | -1.37e+5 | -1.44e+5 | -1.53e+5 | -1.62e+5 |

is stable when the threshold is around 0.3. We have verified that there are similar conclusions on other data sets. Therefore 0.3 should be a reasonable threshold.

## D.3 ML4C: Code and data

For reviewers to check reproducibility of our results reported in §5, we put our code and data on an anonymous site `http://ml4c.xyz`.