# OpenReview forum: "ML4C: Seeing Causality Through Latent Vicinity"
_NeurIPS.cc/2021/Conference — NeurIPS 2021 Submitted_

### Official Review · Reviewer_oFcS · 2021-07-01

**Rating:** 6
**Confidence:** 3

**Summary:**

The paper proposes a novel way of supervised causal learning. The core of the algorithm is a binary classifier that determines if the input UT are in a v structure or not. Given the predicted structures Meek rules are applied to reconstruct the DAG.

**Limitations And Societal Impact:**

A few questions are raised as for limitations of this work:
1) How to determine the sufficient sample size to make a good enough classifier , and if it is possible to do so.
2) What would happen with using different choices on the featurization methods
3) An F1 score of 1 is somewhat suspicious, could the authors please confirm the accuracy of their evaluation. Are the reported numbers the result of multiple trials or the max performance is noted ?
4) Could the authors please put their work into context from a practical point of view?
5) In the case of inference where ground truth is absent, are there ways to evaluate the performance of the algorithm such that checks and balances are introduced to minimize negative societal impact ?

**Main Review:**

Structure and Language: At times the paper lacks clarity and has an unorthodox organisation, for example methodological details are found in the introduction. I would suggest a bit of restructuring and an independent person to read it such that clarity is improved.

Proposed Method: The paper is thorough and the attention to detail is applauded. As above some explanations could be improved in some cases. Line 127 refers to established theory of identifiability, please cite this properly.
In line 239 the authors mention that for a canonical dataset with sufficient samples their method yet there is no mention on how to determine such a quantity, nor if it is possible to do so. It is further claimed that the featurization is not handmade. Technically speaking, even though the features are derived from elaborate heuristics I do not believe it is correct to argue complete independence of features from expert knowledge, as inputs are not derived from raw data. As such a semi-handmade featurization step would be a more appropriate description

Empirical Analysis: The paper offers a quite in depth experimentation/comparison to other methods. Three questions are raised from this analysis.




**Time Spent Reviewing:**

4

---

> ### Author Response · Authors · 2021-08-10
> **Response to Reviewer oFcS**
>
> We appreciate the reviewer's positive feedback and comments. All the comments will be addressed in our revised version. Below are the details:
>
> ---
>
> **(Q1)** Reviewer wonders more detailed analysis on sample size vs. classification accuracy.
>
> In our paper, we mainly prove that (Theorem 1, line 274, bottom of page 6) ML4C converges to a perfect classifier when sample size goes to infinity (note: sample size here refers to the number of records for a given test dataset), our analysis follows the conventional analysis for causal discovery (i.e., analyzing the algorithm's asymptotic correctness), such as those in [2,3,4]. However, we do not provide more detailed analysis on how finite sample size impacts the classification accuracy. Nevertheless, based on our experiment, as shown in Table 3 (top of page 9) on evaluation of robustness, we show that ML4C exhibits good and stable accuracy when sample size varies from 1k to 20k.
>
> In summary, we leave this theoretical analysis as an interesting future work.
>
> ---
>
> **(Q2)** Reviewer wonders model's behavior when choosing difference featurization methods.
>
> The design principle of our featurization comes from two aspects: conditional dependencies and structural entanglement. We provide intuitions (line 194 for conditional dependencies, line 218 for structural entanglement), analysis (lemma 2 and 3) and experiment results to demonstrate the effectiveness of our featurization. We have not considered other aspects. The opportunities of exploiting other featurization aspects to further improve SCL performance is an interesting research direction.
>
> **Remark**: We believe our approach is a first attempt to tackle the problem of SCL systematically, and we foresee quite a few new and interesting research directions. Our response #1 to reviewer [JXwC](https://openreview.net/forum?id=b36m4ZYG1gD&noteId=0mr1KTYTZXd) also discusses one interesting opportunity.
>
> ---
>
> **(Q3)** Reviewer wonders the reproducibility of our results.
>
> Our performance result is highly reproducible. Our classifier is a vanilla XBGoost classifier with fixed hyper-parameters (Experiments/ML4C_accuracy_and_robustness.py L23), thus its output is fully deterministic given a specific input. Our featurization follows deterministic process, as depicted in section 4.2. The only uncertainty comes from kernel mean embedding, where we follow standard approach without any modification. We observed that its impact on perturbation of result is negligible. Please check our code (Tools/Utility.py L66) for details and for reproducibility.
>
> In Table 1 of paper (top of page 8), we report the SHD/F1 based on one normal run. In the table below, we list the average performance with standard deviation based on 5 runs. You can see the result is highly stable. In our revised version, we will use the results based on 5 runs (include both mean and stdv) in Table 1.
>
> | Dataset    	| SHD          	| F1-score    	|
> |------------	|--------------	|-------------	|
> | child      	| 0.0±0.0    	| 1.00±0.00 	|
> | insurance  	| 1.8±1.7    	| 0.95±0.04 	|
> | water      	| 8.6±2.6    	| 0.90±0.03 	|
> | mildew     	| 5.6±1.4    	| 0.88±0.03 	|
> | alarm      	| 1.2±0.4    	| 0.97±0.01 	|
> | barley     	| 3.6±2.4    	| 0.97±0.02 	|
> | hailfinder 	| 11.6±1.0   	| 0.79±0.03 	|
> | hepar2     	| 0.0±0.0    	| 1.00±0.00 	|
> | win95pts   	| 2.0±1.5    	| 0.98±0.01 	|
> | pathfinder 	| 6.8±7.2    	| 0.93±0.06 	|
> | munin1     	| 11.8±1.8   	| 0.96±0.01 	|
> | andes      	| 0.6±0.8    	| 1.00±0.00 	|
> | diabetes   	| 37.6±4.9   	| 0.94±0.01 	|
> | pigs       	| 0.0±0.0    	| 1.00±0.00 	|
> | link       	| 1.6±1.6    	| 1.00±0.00 	|
> | munin      	| 69.2±7.4   	| 0.95±0.00 	|
> | munin2     	| 114.6±27.5 	| 0.92±0.02 	|
> | munin3     	| 104.6±26.2 	| 0.93±0.02 	|
> | munin4     	| 95.8±21.7  	| 0.94±0.01 	|
>
> ---
>
> **(Q4&Q5)** Reviewer wonders more elaboration of adopting ML4C in practice. In particular, when ground truth is absent, how to evaluate the performance of ML4C to mitigate negative societal impact?
>
> We assume "ground truth is absent" means "ground truth skeleton is absent", and we have the following response. ML4C requires accurate skeleton as input, thus identifying an accurate skeleton is one future work (line 357~359) to make ML4C practical. Recently, there have been some novel skeleton learning algorithms such as [5] which could provide accurate skeleton results. Thus, combining ML4C with some novel skeleton learning algorithm is a promising direction we will work on.
>
> That being said, when ML4C's output contains mistakes (e.g., a certain causal direction is incorrect), which could be either due to the mistake of the input skeleton (the edge should not exist) or due to the mistake of ML4C's orientation (the direction is incorrect). It is possible to distinguish these two types of mistakes. According to the definition of skeleton (definition 3, line 138), it is a pure statistical concept without any assumptions. Each edge in the skeleton can be verified by carefully inspecting every possible conditional independence relationships (once there is, the edge should not exist). By excluding the mistakes made from input skeleton , the other type of mistakes made by ML4C are either due to the violation of assumptions (e.g., we assume faithfulness and causal sufficiency) or due to wrong classification of UTs., the performance of ML4C can be further evaluated according to the domain knowledge on some specific cause-effect relationships in the data, or can be further verified by conducting plausible randomized control experiment.
>
> We will add such discussion in our revised version.
>
> ---
>
> **Other comments:**
>
> i) Reviewer suggests that some part of the paper can be better re-structured or refined.
>
> We will add reference for established theory of identifiability on discrete data in line 127, and we will improve our introduction to be better structured in revised version.
>
> ii) Reviewer concerns our claim on "we avoid the need of hand-crafted features".
>
> Well received and agree. The principle of our featurization follows two aspects (conditional dependency and structural entanglement) which are derived from expert knowledge. Thus, it is inappropriate to say that our featurization completely avoids the need of hand-crafted features. We will refine this part, to use more appropriate terms to describe our featurization in our revised version.
>
> ---
>
> [1] Christopher Meek. Strong completeness and faithfulness in bayesian networks. arXiv preprint arXiv:1302.4973, 2013
>
> [2] Gao, T., Fadnis, K. and Campbell, M., 2017, July. Local-to-global Bayesian network structure learning. In International Conference on Machine Learning (pp. 1193-1202). PMLR.
>
> [3] Ramsey, J., Zhang, J. and Spirtes, P.L., 2012. Adjacency-faithfulness and conservative causal inference. arXiv preprint arXiv:1206.6843.
>
> [4] Tsamardinos, I., Brown, L.E. and Aliferis, C.F., 2006. The max-min hill-climbing Bayesian network structure learning algorithm. Machine learning, 65(1), pp.31-78.
>
> [5] Ding, R., Liu, Y., Tian, J., Fu, Z., Han, S. and Zhang, D., 2020, April. Reliable and Efficient Anytime Skeleton Learning. In Proceedings of the AAAI Conference on Artificial Intelligence (Vol. 34, No. 06, pp. 10101-10109).

---

> > ### Comment · Reviewer_oFcS · 2021-08-17
> > **Rebuttal**
> >
> > I thank the authors for their rebuttal, I believe my main questions have been addressed provided the changes promised take place.
> > I will maintain my acceptance recommendation

---

### Official Review · Reviewer_JXwC · 2021-07-16

**Rating:** 7
**Confidence:** 2

**Summary:**

Given a discrete dataset and a graph skeleton consistent with the underlying DAG generating the dataset, this paper proposed a new supervised causal learning (SCL) approach, called ML4C, to classify each unshielded triple in the skeleton as V structure (positive class) or not (negative class).

In the first step, a feature vector is constructed for each unshielded triple X-T-Y, where 1) some features characterize  the conditional dependency between X and Y given T since the value could vary depending on if the triple is truly a V structure; 2) some features characterizes the entanglement between the parent-children set of each node in the triple as well as the sepsets. The extracted features vector is then embedded as a fixed-dimensional vector through kernel mean embedding. In the second step, a classification model is trained to predict the V structure label given the input feature vector.

Experimental results on benchmark datasets show the proposed ML4C outperforms its competing alternatives, including both supervised and unsupervised causal learning approaches, in identifying the ground truth DAG generating the dataset.

**Ethics Review Area:**

["I don’t know"]

**Limitations And Societal Impact:**

The need to use the ground-truth skeleton is the major limitation. I didn't see any negative societal impact.

**Main Review:**

The feature construction for unshielded triple seems novel. The background work is well introduced. Empirical results show strong performance of the proposed approach.

Why does the proposed approach only apply to discrete data? Is there any problem in applying it to continuous data?

Since the ground-truth skeleton is given as input, it would be useful to evaluate how the model performance changes as the input skleleton is misspecified. Furthermore, when comparing against those baseline approaches without skeleton input, it would be more appropriate to start from the same skeleton that is identified by each approach. In this way, the comparison becomes fairer to the baselines.

When evaluating the robustness of the proposed approach, instead of running test on datasets with varying sample size, it would be more interesting to train the model with varying number of DAGs used to train the model. Specifically, what's the minimal number of input DAGs that are needed to reach the best performance?

**Time Spent Reviewing:**

2

---

> ### Author Response · Authors · 2021-08-10
> **Response to Reviewer JXwC**
>
> We appreciate the reviewer's comments and suggestions. All the comments will be addressed in our revised version. Below are the details:
>
> ---
>
> **(Q1)** Reviewer wonders whether our approach can be applied to continuous data.
>
> Operationally, our approach can be easily extended to work on continuous data since there are standard algorithms in literature (e.g., Hilbert-Schmidt Independence Criterion [4]) to be used to calculate conditional dependencies over numerical variables. However, the theory of causal structure identifiability on continuous data can be different. According to the theorem of Markov completeness [1], for discrete data, we can only identify a causal graph up to its Markov equivalence class (i.e., represented by CPDAG). But for continuous data, if it further satisfies additive noise model [2], or linear SEM with non-Gaussian noise [3], some undirected edges in the CPDAG can further be oriented. As a result, on continuous data, the two learning phases (in the two-phase learning paradigm which we have advocated) may not be easily facilitated by a single learning task (i.e., v-structures are no longer sufficient for identifiability), and there's needs to design the corresponding learning tasks with more considerations. In summary, we believe the two-phase learning paradigm sheds light on fundamental question of SCL, it shows an appropriate aspect to conduct SCL on continuous data, which is an open area in literature. We will add this as part of discussion in our revised version.
>
>
> **Remark**: We believe our approach is a first attempt to tackle the problem of SCL systematically, and we foresee quite a few new and interesting research directions. Our response to Q1 and Q2 of reviewer [oFcS](https://openreview.net/forum?id=b36m4ZYG1gD&noteId=gSPHvQSrjR) also discusses two interesting opportunities.
>
> ---
>
> **(Q2.1)** Reviewer suggests adding experiments when ground-truth skeleton is mis-specified.
>
> We thank reviewer for the feedback. We have done this part of experiment informally, and we observed that ML4C consistently achieves top-ranked performance when comparing with other competitors by randomly perturbing some edges in the skeleton. We will conduct systematic experiment (e.g., randomly perturbing 5%, 10%, 20% edges in the skeleton) to evaluate the tolerance of ML4C as well as other competitors and report the results in revised version.
>
>
> ---
>
> **(Q2.2)** Reviewer suggests fairer experiment when comparing with algorithms without skeleton as input.
>
> Below we report a fairer comparison with BLIP [5]. BLIP is the strongest competitor among the three algorithms which do not take skeleton as input (DAG-GNN, BLIP and GOBNILP). Specifically, we follow reviewer's suggestion, to use the skeleton identified by BLIP (i.e., we convert its output from a DAG to the corresponding skeleton) as input for ML4C, and we compare the accuracy of ML4C's output against BLIP's output.
>
> The result is shown in the below table. Among the 23 datasets, ML4C is better than BLIP on 16 datasets, both have same accuracy on 4 datasets and BLIP is better than ML4C only on 3 datasets hailfinder, hepar2 and pigs (even for these datasets, ML4C still has very close accuracy to BLIP). Moreover, for the datasets where BLIP produces skeletons with very low accuracy (such as munin 1~4, skeleton accuracies are around 0.5), ML4C has significantly better accuracy than BLIP, which shows ML4C's has better ability for orientation and also partly shows ML4C's tolerance w.r.t. skeleton perturbations.
>
> | dataset     	| skeleton accuracy (F1) 	| method  	| F1   	| SHD  	| Who is better 	|
> |-------------	|------------------------	|---------	|------	|------	|---------------	|
> |    asia     	|          0.82          	|  BLIP   	| 0.57 	| 6    	|      Tie      	|
> |             	|                        	|  ML4C   	| 0.57 	| 6    	|               	|
> |   cancer    	|          1.00          	|  BLIP   	| 0.00 	| 4    	|      ML4C     	|
> |             	|                        	|  ML4C   	| **1.00** 	| 0    	|               	|
> | earthquake  	|          0.89          	|  BLIP   	| 0.00 	| 5    	|      Tie      	|
> |             	|                        	|  ML4C   	| 0.00 	| 5    	|               	|
> |    sachs    	|          0.97          	|  BLIP   	| 0.00 	| 1    	|      Tie      	|
> |             	|                        	|  ML4C   	| 0.00 	| 14   	|               	|
> |   survey    	|          0.91          	|  BLIP   	| 0.00 	| 6    	|      ML4C     	|
> |             	|                        	|  ML4C   	| **0.73** 	| 2    	|               	|
> |    alarm    	|          0.91          	|  BLIP   	| 0.82 	| 17   	|      ML4C     	|
> |             	|                        	|  ML4C   	| **0.84** 	| 13   	|               	|
> |   barley    	|          0.70          	|  BLIP   	| 0.48 	| 60   	|      ML4C     	|
> |             	|                        	|  ML4C   	| **0.57** 	| 52   	|               	|
> |    child    	|          1.00          	|  BLIP   	| 1.00 	| 0    	|      Tie      	|
> |             	|                        	|  ML4C   	| 1.00 	| 0    	|               	|
> |  insurance  	|          0.78          	|  BLIP   	| 0.51 	| 35   	|      ML4C     	|
> |             	|                        	|  ML4C   	| **0.59** 	| 31   	|               	|
> |   mildew    	|          0.69          	|  BLIP   	| 0.41 	| 36   	|      ML4C     	|
> |             	|                        	|  ML4C   	| **0.56** 	| 31   	|               	|
> |    water    	|          0.48          	|  BLIP   	| 0.20 	| 65   	|      ML4C     	|
> |             	|                        	|  ML4C   	| **0.25** 	| 63   	|               	|
> | hailfinder  	|          0.16          	|  BLIP   	| **0.18** 	| 111  	|      BLIP     	|
> |             	|                        	|  ML4C   	| 0.17 	| 111  	|               	|
> |   hepar2    	|          0.71          	|  BLIP   	| **0.54** 	| 79   	|      BLIP     	|
> |             	|                        	|  ML4C   	| 0.46 	| 85   	|               	|
> |  win95pts   	|          0.71          	|  BLIP   	| 0.47 	| 103  	|      ML4C     	|
> |             	|                        	|  ML4C   	| **0.63** 	| 83   	|               	|
> |    andes    	|          0.80          	|  BLIP   	| 0.76 	| 175  	|      ML4C     	|
> |             	|                        	|  ML4C   	| **0.78** 	| 158  	|               	|
> |  diabetes   	|          0.66          	|  BLIP   	| 0.43 	| 534  	|      ML4C     	|
> |             	|                        	|  ML4C   	| **0.44** 	| 522  	|               	|
> |    link     	|          0.61          	|  BLIP   	| 0.49 	| 947  	|      ML4C     	|
> |             	|                        	|  ML4C   	| **0.53** 	| 916  	|               	|
> |   munin1    	|          0.57          	|  BLIP   	| 0.42 	| 257  	|      ML4C     	|
> |             	|                        	|  ML4C   	| **0.49** 	| 249  	|               	|
> | pathfinder  	|          0.35          	|  BLIP   	| 0.07 	| 241  	|      ML4C     	|
> |             	|                        	|  ML4C   	| **0.12** 	| 259  	|               	|
> |    pigs     	|          1.00          	|  BLIP   	| 1.00 	| 6    	|      BLIP     	|
> |             	|                        	|  ML4C   	| 0.99 	| 12   	|               	|
> |    munin    	|          0.50          	|  BLIP   	| 0.29 	| 1599 	|      ML4C     	|
> |             	|                        	|  ML4C   	| **0.36** 	| 1484 	|               	|
> |   munin3    	|          0.49          	|  BLIP   	| 0.26 	| 1539 	|      ML4C     	|
> |             	|                        	|  ML4C   	| **0.43** 	| 1410 	|               	|
> |   munin4    	|          0.45          	|  BLIP   	| 0.28 	| 1627 	|      ML4C     	|
> |             	|                        	|  ML4C   	| **0.37** 	| 1565 	|               	|             	|
>
> We will add this part of results in our revised version, to make the comparison with no-skeleton-input algorithms fairer.
>
> ---
>
> **(Q3)** Reviewer wonders training size vs. performance on robustness evaluation.
>
> One benefit of supervised causal learning is that obtaining training set is cheap and straightforward since we can sample graphs from DAG space and generate corresponding datasets. Therefore, we could generate as many training instances (note: each training instance here refers to a dataset with ground-truth DAG as label) as we want to push the learner's performance. In our experiment, we use mild scale of training data: 400 graphs, with ~100,000 v-structures (label = 1) and ~300,000 non-v-structures (label = 0) and we already can achieve good results. In fact, from robustness perspective, we find it is interesting to see if ML4C-Learner performs well when it is trained on datasets with fixed sample size (sample size here refers to number of records for each dataset, e.g., 10k) but is tested on dataset with other sample sizes (e.g., 1k). Classical causal learning algorithms use fixed threshold for conditional independence test (e.g., α=0.05 or 0.01), which may error-prone on such setting. Our results on Table 3 (top of page 9) shows that ML4C exhibits consistently good performance which indicates that it learns some more complex and stable decision mechanism.
>
> Reviewer's suggestion is also a very interesting perspective. We also have conducted experiments with different training size and ML4C-Learner exhibits stable performance even when training size is 40 times smaller than current training size (we did not include this part yet). In our revised version, we will add this experiment results as part of robustness evaluation.
>
> ---
>
> Due to characters limit, references will be put in a reply to this comment.

---

> > ### Author Response · Authors · 2021-08-10
> > **Response to Reviewer JXwC cont'd: References**
> >
> > [1] Christopher Meek. Strong completeness and faithfulness in bayesian networks. arXiv preprint arXiv:1302.4973, 2013
> >
> > [2] Patrik Hoyer, Dominik Janzing, Joris M Mooij, Jonas Peters, and Bernhard Schölkopf. Nonlinear causal discovery with additive noise models. Advances in neural information processing systems, 2008
> >
> > [3] Shohei Shimizu, Patrik O Hoyer, Aapo Hyvärinen, Antti Kerminen, and Michael Jordan. A linear non-gaussian acyclic model for causal discovery. Journal of Machine Learning Research, 2006
> >
> > [4] A. Gretton, K. Fukumizu, C. H. Teo, L. Song, B. Sch¨olkopf, and A. Smola. A kernel statistical test of independence. In Advances in Neural Information Processing Systems 20 (NIPS), pages 585–592. MIT Press, 2008.
> >
> > [5] Scanagatta, M., de Campos, C.P., Corani, G. and Zaffalon, M., 2015. Learning Bayesian networks with thousands of variables. In Advances in neural information processing systems (pp. 1864-1872).

---

> > > ### Comment · Reviewer_JXwC · 2021-08-16
> > > **Response to rebuttal**
> > >
> > > Thank the authors for addressing my comments in detail and running extra experiments in such a short time. I have no further questions and will recommend the acceptance of this work.

---

### Official Review · Reviewer_fW1y · 2021-07-16

**Rating:** 6
**Confidence:** 3

**Summary:**

This paper aims to tackle the supervised causal learning (SCL) problem, i.e., learning causal relationships from observational data in a supervised manner. The paper first shows that SCL is beneficial only when the learning target is identifiable. So this paper proposes a two-phase paradigm which first classifies if the target is identifiable and then orients the unshielded triple (UT). To achieve both objectives, the core of this paper is to train a classier to determine if a given UT constitutes a collider. The authors utilize the intuition that stronger entanglement implies smaller chance of identifiability and propose to make use of the overlap between the vicinties. Theoretical and empirical analysis both demonstrate the effectiveness of the proposed framework.

**Limitations And Societal Impact:**

Please kindly see the question in the above section.

**Main Review:**

**Originality**: This paper proposes an interesting approach for solving the SCL problem, which breaks it down to classifying the identifiability of UTs.

**Quality**: This paper is overall technically sound with strong empirical evaluation results demonstrated. Notations are clearly defined and theoretical analyses are also carried out.

**Clarity**: The paper is generally well-written and organized. However, some details are not clear to me; please see the questions below.

**Significance**: The target problem of this paper, SCL, is an important and fundamental task to be addressed, and it appears to me that the approach presented in this paper can advance the state of the art in solving SCL problem.

**My questions**:
1. How are the embeddings of each UT computed? what is the dimension?
2. The paper utilizes the intuition that stronger entanglement implies smaller chance of identifiability, can this be proved?
3. If this intuition holds strictly, can we simply fit a linear regression using the overlap coefficient as mentioned in line 226 to determine the dependency? why a classifier is needed?
4. From Table 1, it seems that NCC performs much better for smaller datasets and ML4C is significantly better for larger datasets. As both are supervised methods, what makes ML4C worse than NCC with small datasets?

---
**Post rebuttal**
Thanks for the responses from the authors. My concerns are mostly resolved. Please incorporate the responses in the new version of the paper.

**Time Spent Reviewing:**

4

---

> ### Author Response · Authors · 2021-08-10
> **Response to Reviewer fW1y**
>
> We appreciate the reviewer for the positive feedback to our work. All the comments will be addressed in our revised version. Below are the details:
>
> ---
>
> **(Q1)** Reviewer wonders more details of computing kernel mean embedding in our approach, and the dimensionality of feature vector.
>
> Dimensionality of feature vector is 755. Specifically, to represent each extended conditional dependency $D=\mathbf{A}\sim\mathbf{B}\vert \mathcal{Z}$ (i.e., a set of scalars with varied set size, line 190), we use standard kernel mean embedding technique in [1] to obtain the embedded vector $ \frac { 1} { | D | } \sum _ { z \in D } \left( \cos \left( \left\langle w _ { j } , z \right\rangle + b _ { j } \right) \right) _ { j = 1 } ^ { m } \in \mathbb { R } ^ { m }$. Here m=15, which means embedding dimensionality for each extended conditional dependency is 15. We also include 5 additional statistics (max, min, mean, std, and set size). Besides, we use both p-value and severity (section B.1 in appendix) to quantify each extended conditional dependency separately. Of the 4 bivariable × 5 conditional = 20 pairs, only one (X;Y|T) is unitary (i.e., it is a single scalar), so we apply embedding to the rest 19 pairs. In addition, entanglement is considered in terms of 5 scaling and 7 overlap coefficients. Thus, the total dimensionality of feature vector is 755 = 5 (scaling) + 7 (overlaps) + 2 * 1 (unitary) + [1 (set size) + 2 * 4 (mean/std/max/min) + 2 * 15 (embedding dimensionality)] * 19 (pairs). See our code at [http://ml4c.xyz](http://ml4c.xyz) for more details at implementation level (Tools/Utility.py L61, BayesianNetwork/CITester.py L104, Experiments/GenerateFeatures.py L121). We will add the details in revised version (mainly in the appendix due to page limit).
>
> ---
>
> **(Q2 & Q3)** Reviewer wonders whether the intuition about "stronger entanglement implies smaller chance of identifiability" can be strictly proved. If so, reviewer wonders if our sophisticated ML model (i.e., XGBoost) is necessary (rather than simple ML model such as linear/logistic regression) to detect v-structures.
>
> In our approach, both conditional dependencies and structural entanglement are key aspects for featurization, which leads to weak or strong discriminative predicates (see lemma 2 and 3 for some examples, line 253-261). Regarding structural entanglement, it is highly relevant to detecting v-structures, for example, $OLP(T,\mathcal{S})=0$ in lemma 3 (line 259) is a strong predicate which is derived from structural entanglement aspect. Such features greatly help ML4C-Learner to achieve high accuracy. However, it is not our intention to make a monotonic claim: "stronger entanglement implies smaller chance of identifiability". Mathematically, this is a non-trivial conjecture and we have not verified it yet. In our revision, we will refine the intuition of entanglement (line 218-220), to void confusion on such 'monotonic' relationship between entanglement and identifiability.
> Lastly, Table 2 (above line 329) shows the power of adopting sophisticated ML models (e.g., XGBoost is an example of sophisticated ML model). We compared ML4C with 4 strong discriminative predicates. Each predicate is a human-specified logic and it can be viewed as a naïve classifier (line 250, 251) which can achieve perfect performance under ideal setting (e.g., sample size goes to infinite, correct conditional independence tests). In practice we are facing non-ideal settings, thus sophisticated ML models leverage supervision to learn complex decision boundaries, which are less error-prone than human-specified logics thus achieve better performance.
> We will add this discussion in our revised version, in "reliability" section (line 338) in experiment.
>
> ---
>
> **(Q4)** Reviewer wonders why ML4C performs worse than NCC on small datasets.
>
> Below we show that ML4C still performs better than NCC on small datasets, in the sense that ML4C exhibits stable and top-ranked performance, while NCC's performance fluctuates drastically regardless of data scale. Specifically, we separate datasets into 2 categories: small (#nodes<=100, win95pts and above) and big (#nodes>100, pathfinder and below).
>
> 1. **Overall comparison**. We see that NCC ranks #1 on 8 datasets, while ML4C ranks #1 on 11 datasets. Within the 8 datasets where NCC ranks first (F1-score=1.00), ML4C performs quite close to NCC (ranks 2.0±0.7, F1-score=0.93±0.08). But within the 11 datasets where ML4C ranks first (F1-score=0.96±0.04), NCC performs significantly worse (ranks 7.0±3.9, F1-score=0.49±0.30).
> 2. **On small datasets**. The basic statistics are as follows: Ranking: ML4C ranks 1.8±0.7, NCC ranks 4.8±4.6; F1-score, ML4C=0.94±0.06, NCC=0.72±0.33. In fact, NCC's performance fluctuates on small datasets, e.g., NCC's F1-scores on child (the smallest dataset in Table 1) is only 0.12.
> 3. **On big datasets**. The basic statistics are as follows: Ranking, ML4C ranks 1.3±0.5, NCC ranks 5.3±3.7. F1-score, ML4C=0.94±0.07, NCC=0.62±0.34.
>
> In summary, the statistics show that ML4C exhibits stable and significantly better performance regardless of data scale. We will add more illustration of this in revised version.
>
> ---
>
> [1] Alex Smola, Arthur Gretton, Le Song, and Bernhard Schölkopf. A hilbert space embedding for distributions. In International Conference on Algorithmic Learning Theory, pages 13–31. Springer, 2007

---

> > ### Comment · Reviewer_fW1y · 2021-08-29
> > **Thanks for the response**
> >
> > I would like to thank the authors for the detailed response. My concerns are mostly resolved. Please incorporate the responses in the new version of the paper.

---

> ### Comment · Area_Chair_42ft · 2021-09-01
> **additional reference**
>
> just a minor remark: this paper seems to be relevant too:
> https://proceedings.mlr.press/v37/lopez-paz15.html

---

> > ### Author Response · Authors · 2021-09-01
> > **response to AC**
> >
> > We thank AC for the kind reminder, and we will add the suggested reference [1] accordingly. In fact, we are well aware of the RCC work by two parts: [1] is the theoretical version (which you suggest to cite), and [2] is the algorithmic version (which has already been cited as [23] in paper). In the related work section, we meant to relate ML4C with [1] (line 98 and line 103 ~ 109), and in the experiment section, we compare ML4C with [2] (line 289). So it was our mistake to miss reference [1].
> > In revised version, we will add [1] in the related work section.
> >
> > [1] Lopez-Paz, D., Muandet, K., Schölkopf, B. and Tolstikhin, I., 2015, June. Towards a learning theory of cause-effect inference. In International Conference on Machine Learning (pp. 1452-1461). PMLR.
> >
> > [2] Lopez-Paz, D., Muandet, K. and Recht, B., 2015. The Randomized Causation Coefficient. J. Mach. Learn. Res., 16, pp.2901-2907.

---

### Decision · Program_Chairs · 2021-09-27

**Decision:**

Reject

**Comment:**

Although reviewers found the idea intriguing, our discussions raised quite some concerns regarding sloppiness of presentation, particularly in the theory part. Just a few examples:

- Without further specification, Proposition 1 is a pretty vague statement.

- There were also the question what the non-trivial content of Proposition 2 is supposed to be

- The statement 'skeleton is a statistical concept'. If we are talking about the skeleton of the causal DAG, the skeleton is also causal. The relation between conditional independences and skeleton is  subject to faithfulness, i.e., a postulate relating statistics to causality.

[Update] Dear authors,

I understand your disappointment on the decision, given that the scores qualify this paper as close to acceptance, I also understand the feeling of unfair treatment. Before I go into a more detailed discussion, I would like to emphasize that the yes/no decision accept/reject is not a decision on good versus bad, often there is a large grey zone of close to acceptance, in which this paper definitely is.

Since the reviewer with score 7 had low confidence, we were left with 2 times the score ‘marginally above the threshold’, which expresses some hesitation after all. After reading the paper myself, I largely agreed with these assessments, but with a *slightly* more critical opinion, and all the remarks below only justify rejection in the context of very high standards of NeurIPS (although I do know that also bad papers can make it sometimes to NeurIPS).

- Proposition 1: it seems highly misleading to me (even without asking for mathematical rigor) for the following reason. Identifiability is a purely theoretical construction that depends on the model assumptions. While traditional causal discovery uses nothing beyond faithfulness, further model assumptions like additive noise render additional directions identifiable). I know that the authors are aware of this, but in real life we have never hard restrictions on the model class. Instead, some patterns are more likely to occur in one direction than the other. Supervised causal discovery has the potential to cope with these missing hard constraints and get evidence for one direction versus the other without a priori given hard constraints on the model classes. In other words, identifying directions without hard criteria is exactly what I would expect from the method.
- Regarding proposition 2, I can see its justification from the context, but phrasing it as a proposition pretends mathematical rigor that I cannot see. What are the assumptions here? E.g. it assumes correctness of the skeleton in the sense of causal semantics (i.e. when faithfulness is met), right?
- I can see now the author’s viewpoint on the skeleton being ‘statistical’ in the context of how they introduced it. I’m sorry about this miss!

As I said, I admit that all these points can justify rejection only in light of very high standards. I really encourage the authors to resubmit! I have to say that I spend quite some time with this decision because of the large grey zone of borderline papers. I should have been more explicit about this in my meta-review – also to make clear that this rejection was not based on a quick ad-hoc decision.

There is no question that other ACs could have come to a different conclusion.  I should also say that I have no reservations per se against any methods of this paper.

Further, I admit that I could have raised the above points during the discussion and the main justification for this miss is the work load of a bunch of other papers that deserved my attention too. I hope these remarks make it easier to just accept the decision as a result of a process with uncertainty, and without absolute truth.

Best wishes,

Area Chair